# Continuous Reconstruction of Spatiotemporal Dynamical System with Neural PDE

## Abstract

Recent studies have shown great promise in applying neural ordinary differential equations (ODEs) for spatiotemporal dynamical systems reconstruction, and the continuous modeling capability is naturally applied to capture the temporal evolution. However, we notice that few studies have employed ODE for spatial relationship modeling. Most of them utilized discrete models, such as CNNs or GNNs, ignoring the inherent continuity in spatial domain. To fill this gap, we propose Neural Partial Differential Equation (Neural PDE) to model spatiotemporal dynamical systems continuously and simultaneously by fitting the higher-order partial derivatives of the hidden states with respect to the spatial and temporal dimensions. Meanwhile, it is combined with a generative pretraining framework to dynamically optimize the initial latent states using an auto-decoder. For efficient training, we propose the overlapping multiple shooting, which splits long trajectories into overlapping sub-segments and enforces smooth transitions using continuity loss. Experiments show that Neural PDE achieves SOTA performance on both advection equation and Burgers equation, e.g., 74.29% RMSE reduction compared to the second-best method under 1% extremely sparse observations, and performs robustly in different resolutions and tasks generalization, which verifies the superiority of the continuous modeling and generalization capabilities.

## 1 Introduction

Spatiotemporal dynamical systems are widely encountered in applications such as fluid dynamics, geophysics, astrophysics, and atmospheric physics. In these contexts, having access to the full spatiotemporal field data is essential for accurately understanding and controlling the underlying processes. However, in practice, the available sensor data is often sparse, with a limited number of sensors failing to capture the complexity of the system dynamics. This makes the task of reconstructing the complete spatiotemporal dynamics from sparse observations both crucial and challenging.

**Related Work.** Existing data-driven methods can be divided into classical and neural network-based approaches. Early works mainly used techniques such as proper orthogonal decomposition (POD) (Kondrashov & Ghil, 2006), dynamic mode decomposition (DMD) (Schmid, 2010), and Galerkin transforms (Boisson & Dubrulle, 2011) for reconstruction tasks. These methods generally rely on simplifying assumptions and linear representations to extract features from complex systems. Therefore, they often struggled with the inherent complexity and nonlinearity of spatiotemporal dynamic systems, limiting their practical applicability.

Deep learning has recently attracted attention for reconstructing spatiotemporal dynamics. Many existing methods use Convolutional Neural Networks (CNNs) (Long et al., 2018; 2019; Ayed et al., 2020; Chai et al., 2020) or Graph Neural Networks (GNNs) (Lienen & Günnemann, 2022) to model temporal evolution. In addition, several diffusion-based methods (Huang et al., 2024; Zhuang et al., 2024; Shu et al., 2023; Li et al., 2024) adopt CNNs as their denoising backbone. For example, $S^3GM$ (Li et al., 2024) leverages a score-based generative framework combined with a Video U-Net model to reconstruct full-field spatiotemporal dynamics. However, a key limitation of these discretized models is their dependence on a fixed resolution, which leads to performance degradation when tested on resolutions different from the training resolution. Specifically, as the input resolution increases, the size of each grid cell decreases, which reduces the actual receptive field of each CNN unit.

In contrast to discretized models such as CNNs and GNNs, some methods model space continuously. Operator-based models (Kovachki et al., 2023), such as the Fourier Neural Operator (FNO) (Li et al., 2021) and Laplace Neural Operator (LNO) (Cao et al., 2024), learn mappings between functions, extending neural networks to infinite-dimensional function spaces. Although these methods model space continuously, the temporal dimension is still discretized because they are autoregressive (Takamoto et al., 2022). In contrast, coordinate-based models, called Implicit Neural Representations (INRs) (Sitzmann et al., 2020), take both spatial and temporal coordinates as input, enabling spatiotemporal continuity. Examples include Physics-Informed Neural Networks (PINNs) (Raissi et al., 2019), which solve PDEs using physical laws but require retraining when conditions change, and MMGNet (Luo et al., 2024), which replaces the time index with a context-aware mechanism, thus losing temporal continuity. Moreover, these end-to-end models are trained for specific tasks and fail to generalize to unseen tasks.

Since spatiotemporal dynamical systems are inherently continuous, motivated by the potential of Neural ODEs (Chen et al., 2018) for modeling continuous dynamics, we explore their use for reconstructing spatiotemporal dynamics. However, directly applying Neural ODEs to spatiotemporal data presents three significant challenges. First, Neural ODEs are designed to model continuous dynamical systems, where the state evolves continuously along a specific dimension. To apply Neural ODEs to spatiotemporal data, a natural approach is to model each dimension with a separate Neural ODE. However, this approach inherently produces a non-conservative vector field, violating path independence. This inconsistency conflicts with the fundamental properties of spatiotemporal dynamical systems, making direct application infeasible. A detailed discussion is provided in Section 3.1. Second, Neural ODEs rely solely on initial observations, which limits their ability to model complex dynamics. Third, the sequential nature of Neural ODEs leads to long training times (Ribeiro et al., 2020; Metz et al., 2021).

**Contribution.** To tackle these challenges, we propose Neural PDE for continuous modeling of spatiotemporal dynamics, as shown in Figure 1. Unlike using separate Neural ODEs for each dimension, which breaks path independence, Neural PDE naturally satisfies this property. Meanwhile, it is combined with a generative pretraining framework. During training, we avoid making task-specific assumptions or learning a direct input-output mapping. Instead, we pretrain the parameters of the Neural PDE on non-sparse data to learn the underlying equations of the spatiotemporal system.

During testing, we optimize the initial latent states based on sparse observations, effectively encoding varying initial conditions, boundary conditions, and equation parameters of the spatiotemporal system. This addresses a key limitation of Neural ODEs, which rely solely on the initial value. Additionally, to overcome the sequential nature of Neural ODEs, we propose overlapping multiple shooting and derive a corresponding loss function with a continuity penalty to enforce smooth transitions between sub-trajectories. Our contributions are:

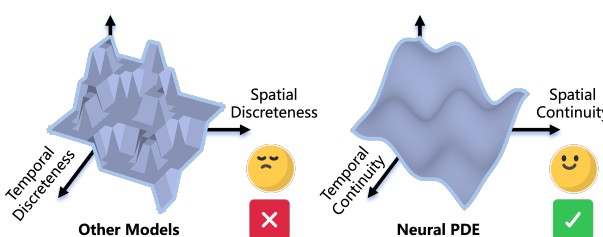

Figure 1: Most existing works exhibit temporal or spatial discreteness, whereas our method preserves temporal and spatial continuity.

- This paper studies how to use neural ODEs to model the dynamics system continuously on both spatial and temporal dimensions with a generative pretraining framework. In contrast, most existing works adopt an end-to-end framework and exhibit temporal or spatial discreteness, rather than modeling both dimensions continuously.

- Correspondingly, we propose Neural PDE, which fits the higher-order partial derivatives of hidden states with respect to spatial and temporal dimensions. This enables reconstruction to be both continuous and simultaneous, avoiding the path-dependence problem where different sequential orders in spatial and temporal modeling lead to inconsistent results.

- We propose overlapping multiple shooting which splits long trajectories into overlapping sub-trajectories to enhances training efficiency and stability. Furthermore, a continuity loss constraint is proposed to enforce smooth transitions between sub-trajectories.

## 2 FORMALIZATION

In this work, we aim to reconstruct full spatiotemporal dynamic systems $\mathcal{D}$. Each trajectory $\mathbf{F} \in \mathcal{D}$ is represented as a tensor of the form $\mathbf{F} \in \mathbb{R}^{T \times N_1 \times N_2 \times \cdots \times N_d \times M}$, where $T$ denotes the number of time steps, $N_i$ represents the number of points along the $i$-th spatial dimension, and $M$ is the number of features. The set of time steps is denoted by $\mathcal{T} = \{t_1, t_2, \ldots, t_T\} \subset \mathbb{R}$, while $\mathcal{X}_1, \mathcal{X}_2, \ldots, \mathcal{X}_d$ denote the sets of points along each spatial dimension, where $\mathcal{X}_i = \{x_{i1}, x_{i2}, \ldots, x_{iN_i}\} \subset \mathbb{R}$ denotes the sets of points along the $i$-th spatial dimension. Both the time steps $\mathcal{T}$ and spatial points $\mathcal{X}_i$ for each $i \in \{1, 2, \ldots, d\}$ are irregular. For each time step $t \in \mathcal{T}$, we define the temporal slice $\mathbf{F}_t \in \mathbb{R}^{N_1 \times N_2 \times \cdots \times N_d \times M}$. The full tensor $\mathbf{F}$ can thus be viewed as a sequence of such temporal slices, i.e., $\mathbf{F} = \{\mathbf{F}_{t_1}, \mathbf{F}_{t_2}, \ldots, \mathbf{F}_{t_T}\}$. We are given sparse measurements of the full spatiotemporal field, which correspond to a set of indices $\mathcal{S} \subset \mathcal{T} \times \mathcal{X}_1 \times \cdots \times \mathcal{X}_d$, where each element $s \in \mathcal{S}$ represents an index corresponding to an observed entry $\mathbf{F}_s \in \mathbb{R}^M$ of the tensor $\mathbf{F}$. Our goal is to learn a function $f$ that reconstructs the complete spatiotemporal dynamic $\mathbf{F}$ from the sparse data.

$$\{\mathbf{F}_s\}_{s \in \mathcal{S}} \xrightarrow{f} \mathbf{F} \tag{1}$$

## 3 METHODOLOGY

In this section, we introduce the proposed method, as shown in Figure 2. The framework begins by solving the Neural PDE using the initial latent states to compute the full latent states. These latent states are subsequently mapped to the output space via an MLP. During training, we pretrain the parameters of the Neural PDE on non-sparse data to learn the underlying equations of the spatiotemporal system. During testing, we optimize the initial latent states based on sparse observations, effectively encoding varying initial conditions, boundary conditions, and equation parameters of the spatiotemporal system. In the following subsections we provide detailed descriptions of the method's five components: Neural PDE, Path Independence Verification, AutoDecoder, Overlapping Multiple Shooting, and the loss function.

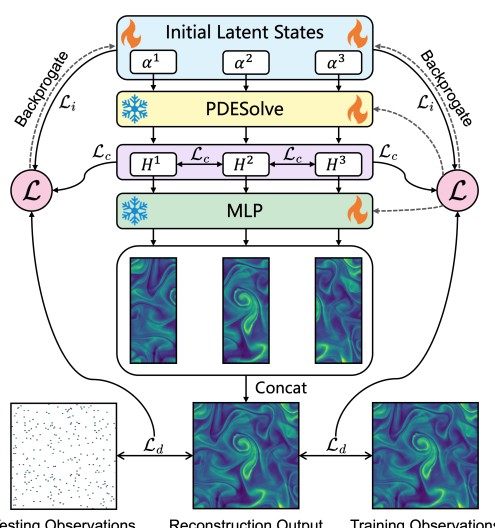

Figure 2: Overview of the proposed methodology.

### 3.1 NEURAL PDE

For a binary function $f(x, y)$, its gradient field $\nabla f(x, y) = \left( \frac{\partial f}{\partial x}, \frac{\partial f}{\partial y} \right)$ is a conservative vector field, which inherently satisfies path independence. When modeling this function using Neural ODEs (Chen et al., 2018), a natural approach is to employ separate Neural ODEs along the $x$- and $y$-directions, respectively:

$$\frac{\partial f}{\partial x}(x, y) = \mathcal{F}_1(x, f(x, y); \theta_1), \tag{2}$$

$$\frac{\partial f}{\partial y}(x, y) = \mathcal{F}_2(y, f(x, y); \theta_2), \tag{3}$$

where $\mathcal{F}_1$ and $\mathcal{F}_2$ are neural networks, such as MLPs, with parameters $\theta_1$ and $\theta_2$, respectively.

**Proposition 3.1.** *The vector field $(\mathcal{F}_1, \mathcal{F}_2)$ is non-conservative, i.e., it does not satisfy path independence.*

The non-conservativeness of the vector field is due to the fact that the neural networks $\mathcal{F}_1$ and $\mathcal{F}_2$ are independent of each other. The two integration paths (first along the $x$-direction, then the $y$-direction, or first along the $y$-direction, then the $x$-direction) lead to different results, thus violating path independence. Proofs are provided in the Appendix B.

To address the problem of Neural ODEs described above, we use neural partial differential equation (Neural PDE) to model $f(x, y)$. For a binary function $f(x, y)$ with gradient field $\nabla f(x, y) = \left( \frac{\partial f}{\partial x}, \frac{\partial f}{\partial y} \right)$. This gradient field is irrotational, implying that its curl vanishes, as expressed by:

$$\frac{\partial}{\partial x} \left( \frac{\partial f}{\partial y} \right) - \frac{\partial}{\partial y} \left( \frac{\partial f}{\partial x} \right) = 0. \tag{4}$$

In other words, the second-order mixed partial derivatives of $f(x, y)$ are commutative:

$$\frac{\partial^2 f}{\partial x \partial y} = \frac{\partial^2 f}{\partial y \partial x}. \tag{5}$$

We introduce a PDE specified by a neural network:

$$\frac{\partial^2 f}{\partial x \partial y}(x, y) = \frac{\partial^2 f}{\partial y \partial x}(x, y) = \mathcal{F}(x, y, f(x, y); \theta), \tag{6}$$

where $\mathcal{F}$ is a neural network, such as an MLP, and $\theta$ denotes its parameters.

By integrating the second-order partial derivatives, we obtain the first-order partial derivatives of the function $f(x, y)$. Specifically, for the partial derivative of $x$, we have:

$$\begin{aligned} \frac{\partial f}{\partial x}(x, y) =& \frac{\partial f}{\partial x}(x, y_0) + \int_{y_0}^{y} \frac{\partial^2 f}{\partial x \partial y}(x, y') dy' \\ =& \mathcal{F}_x(x, y_0, f(x, y_0); \theta_x) + \int_{y_0}^{y} \mathcal{F}(x, y', f(x, y'); \theta) dy', \end{aligned} \tag{7}$$

where the initial value of the first-order partial derivative $\frac{\partial f}{\partial x}(x, y_0)$ is specified by the neural network $\mathcal{F}_x$, such as an MLP, with its own set of learnable parameters $\theta_x$.

Similarly, for the partial derivative of $y$, we have:

$$\begin{aligned} \frac{\partial f}{\partial y}(x, y) =& \frac{\partial f}{\partial y}(x_0, y) + \int_{x_0}^{x} \frac{\partial^2 f}{\partial y \partial x}(x', y) dx' \\ =& \mathcal{F}_y(x_0, y, f(x_0, y); \theta_y) + \int_{x_0}^{x} \mathcal{F}(x', y, f(x', y); \theta) dx', \end{aligned} \tag{8}$$

where $\frac{\partial f}{\partial y}(x_0, y)$ is specified by the neural network $\mathcal{F}_y$ (e.g., MLPs), with learnable parameters $\theta_y$.

**Proposition 3.2.** *The vector field formed by the expressions on the right sides of Equation 7 and Equation 8 is conservative, i.e., it satisfies path independence. The integration result is given by:*

$$\begin{aligned} f(x, y) =& f(x_0, y_0) + \int_{x_0}^{x} \mathcal{F}_x(x', y_0, f(x', y_0); \theta_x) dx' \\ &+ \int_{y_0}^{y} \mathcal{F}_y(x_0, y', f(x_0, y'); \theta_y) dy' + \int_{y_0}^{y} \int_{x_0}^{x} \mathcal{F}(x', y', f(x', y'); \theta) dx' dy'. \end{aligned} \tag{9}$$

Neural PDE is path-independent in modeling $f(x, y)$, making it more advantageous than Neural ODEs (see Appendix B for a detailed proof).

**Proposition 3.3.** *Neural PDE can be solved using nested ODE solvers, with the solution process encapsulated in the function PDESolve:*

$$f(x, y) = PDESolve(f(x_0, y_0), x_0, x, y_0, y). \tag{10}$$

A detailed proof is provided in the Appendix B. This result suggests that methods developed for Neural ODEs can be directly applied to solve Neural PDE. In our experiments, we utilized the standard torchdiffeq (Chen, 2018) package without any modifications. A key benefit of Neural ODEs is the ability to train via adjoint backpropagation, which can also be leveraged for Neural PDE. This approach enables constant space complexity with respect to the depth of the solver, a significant advantage for modeling functions on high-resolution grids where many integration steps

are required. We extend PDESolve from scalar values of $x$ and $y$ to grids $\mathcal{X}$ and $\mathcal{Y}$ by defining the solution over a grid of points:

$$\hat{\mathbf{F}} = \text{PDESolve}(\mathbf{F}_{x_0, y_0}, \mathcal{X}, \mathcal{Y}), \tag{11}$$

where $\hat{\mathbf{F}} \in \mathbb{R}^{N_x \times N_y \times M}$ is the tensor of solutions, with $N_x$ and $N_y$ representing the number of points in $\mathcal{X}$ and $\mathcal{Y}$, respectively. Each element $\hat{\mathbf{F}}_{x_i, y_j}$ corresponds to the solution $f(x_i, y_j) \in \mathbb{R}^M$, where $x_i \in \mathcal{X}$ and $y_j \in \mathcal{Y}$, computed from the initial condition $\mathbf{F}_{x_0, y_0} \in \mathbb{R}^M$ using PDESolve.

Our framework naturally generalizes from the 2D case to higher spatiotemporal dimensions. (see Appendix B for the 3D case). For a system with one temporal and $d$ spatial dimensions, we compute the solution over a discrete grid defined by time points $\mathcal{T}$ and spatial points $\{\mathcal{X}_i\}_{i=1}^d$. Given an initial state $\mathbf{F}_{\text{init}} = f(t_0, \mathbf{x}_0) \in \mathbb{R}^M$, the full solution tensor $\hat{\mathbf{F}}$ is computed as:

$$\hat{\mathbf{F}} = \text{PDESolve}(\mathbf{F}_{\text{init}}, \mathcal{T}, \mathcal{X}_1, \dots, \mathcal{X}_d), \tag{12}$$

where $\hat{\mathbf{F}} \in \mathbb{R}^{N_t \times N_1 \times \cdots \times N_d \times M}$, with $N_t = |\mathcal{T}|$, $N_i = |\mathcal{X}_i|$, and $\mathbf{x}_0 = (x_{1,0}, \dots, x_{d,0})$.

### 3.2 Path Independence Verification

To evaluate the path independence properties of Neural PDE, we consider the following function, although other functions can also be used.

$$f(x, y) = \sin(x)\cos(y), \quad -\pi \le x, y \le \pi. \tag{13}$$

During training, the integration is performed first along $x$, then $y$. During testing, the integration path is reversed: first along $y$, then $x$. By employing different integration paths during training and testing, we can evaluate the path independence of the Neural PDE.

Figure 3 presents the experimental results. Neural PDE demonstrates its capability to accurately model the gradient field of the target function, enabling consistent reconstructions irrespective of the integration path. In contrast, the Neural ODE models the $x$- and $y$-components of the vector field with two independent networks, without enforcing the equality of mixed partial derivatives required for a conservative field. Consequently, the learned field is path-dependent and exhibits poor performance on integration paths different from those used in training.

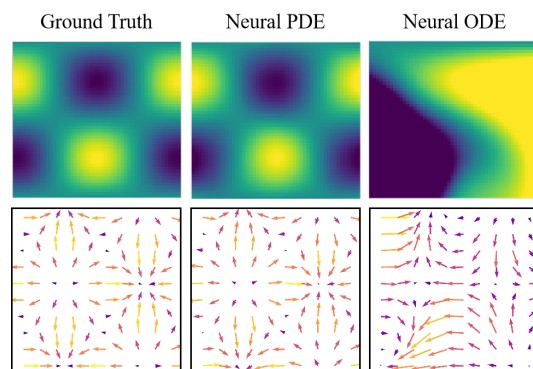

Figure 3: The first row shows function values: the true value on the left, and reconstructions by Neural PDE and Neural ODE in the middle and right. The second row shows gradient fields: the true field on the left, and vector fields modeled by Neural PDE and Neural ODE in the middle and right.

### 3.3 Auto-decoder

Neural PDE shares the same limitation as Neural ODE: the solution is determined solely by the initial conditions, and the trajectory cannot be adapted based on subsequent observations. To address this, we introduce auto-decoding (Park et al., 2019) as a solution.

In the auto-decoder framework, there is no encoder; instead, each trajectory $\mathbf{F}$ is associated with a learnable latent state $\boldsymbol{\alpha}$. During training, both the decoder and the latent state are jointly optimized. During testing, the decoder parameters are fixed, and only the latent state is updated. We use $\boldsymbol{\alpha}$ as the initial state for the Neural PDE, modifying the solution formulation from Equation 12 to:

$$\mathbf{H} = \text{PDESolve}(\boldsymbol{\alpha}, \mathcal{T}, \mathcal{X}_1, \dots, \mathcal{X}_d), \tag{14}$$

where $\mathbf{H} \in \mathbb{R}^{N_t \times N_1 \times \cdots \times N_d \times D}$ is the tensor of solutions, and $\boldsymbol{\alpha} \in \mathbb{R}^D$ is the initial latent state. We then use MLP to map the latent state to the output space:

$$\hat{\mathbf{F}} = \text{MLP}(\mathbf{H}), \tag{15}$$

where $\hat{\mathbf{F}} \in \mathbb{R}^{T \times N_1 \times \cdots \times N_d \times M}$ is the final output.

### 3.4 OVERLAPPING MULTIPLE SHOOTING

Another disadvantage of Neural PDE, similar to Neural ODE, is their slow training time, primarily due to the sequential nature of the integration process. A simple and efficient approach to address this issue is multiple shooting (Van Domselaar & Hemker, 1975; Bock & Plitt, 1984; Iakovlev et al., 2023), which splits a trajectory into short, non-overlapping sub-trajectories that are optimized in parallel. However, this method cannot guarantee smooth continuity, as there is no overlap between sub-trajectories.

To overcome this limitation, we propose overlapping multiple shooting, a method that introduces overlap between adjacent sub-trajectories to improve continuity. Specifically, we split a trajectory $\mathbf{F}$ into $B$ consecutive sub-trajectories $\mathbf{F}^{1:B}$, each with its respective initial latent states $\boldsymbol{\alpha}^{1:B}$. The number of sub-trajectories, $B$, is given by $B = \lceil \frac{T-O}{L-O} \rceil$, where $L$ denotes the length of each sub-trajectory, $O$ is the overlap between adjacent sub-trajectories. The time indices for the $b$-th sub-trajectory, $\mathcal{T}^b$, are given by $\{t_{(b-1)*(L-O)+1}, \ldots, t_{(b-1)*(L-O)+L}\}$, and the $b$-th sub-trajectory is defined as $\mathbf{F}^b = \{\mathbf{F}_i\}_{i \in \mathcal{T}^b}$. For each $b \in \{1, 2, \ldots, B\}$, we solve the Neural PDE:

$$\mathbf{H}^b = \text{PDESolve}(\boldsymbol{\alpha}^b, \mathcal{T}^b, \mathcal{X}_1, \ldots, \mathcal{X}_d), \tag{16}$$

where $\mathbf{H}^b \in \mathbb{R}^{L \times N_1 \times \cdots \times N_d \times D}$ is the solution tensor, and $\boldsymbol{\alpha}^b \in \mathbb{R}^D$ is the initial latent state of the $b$-th sub-trajectory. An MLP then maps the latent state to the output space:

$$\hat{\mathbf{F}}^b = \text{MLP}(\mathbf{H}^b), \tag{17}$$

where $\hat{\mathbf{F}}^b \in \mathbb{R}^{L \times N_1 \times \cdots \times N_d \times M}$.

Finally, the sub-trajectories are concatenated to form the final reconstructed output $\hat{\mathbf{F}}$. For each time index $t_i \in \mathcal{T}$ with sub-trajectory index $b$, the output at $t_i$ is:

$$\hat{\mathbf{F}}_{t_i} = \hat{\mathbf{F}}_{t_i}^b. \tag{18}$$

For simplicity, the overlapping values are from the previous sub-trajectory. The loss function enforces consistency of the overlapping values between adjacent sub-trajectories.

### 3.5 LOSS FUNCTION

To estimate the parameters $\theta$ of the Neural PDE and the initial latent states $\boldsymbol{\alpha}_{\mathbf{F}}^{1:B}$ for each trajectory $\mathbf{F}$, we adopt a probabilistic framework. We begin by defining the continuity inducing prior as follows:

$$p_\theta(\boldsymbol{\alpha}_{\mathbf{F}}^{1:B}) = p(\boldsymbol{\alpha}_{\mathbf{F}}^1) \prod_{b=2}^{B} p_\theta(\boldsymbol{\alpha}_{\mathbf{F}}^b | \boldsymbol{\alpha}_{\mathbf{F}}^{b-1}) = \mathcal{N}(\boldsymbol{\alpha}_{\mathbf{F}}^1 | 0, \sigma_i^2 I) \prod_{b=2}^{B} \mathcal{N}(\boldsymbol{\alpha}_{\mathbf{F}}^b | \mathbf{H}_{idx}^{b-1}, \sigma_c^2 I), \tag{19}$$

where the initial state $\boldsymbol{\alpha}_{\mathbf{F}}^1$ follows a zero-mean Gaussian distribution with covariance $\sigma_i^2 I$ and the initial state of each sub-trajectory is conditioned on the previous one. Here, $I$ is the identity matrix, and $\sigma_i$ and $\sigma_c$ control regularization strength for the initial state and continuity. $\mathbf{H}_{idx}^{b-1}$ is computed by solving the Neural PDE at the previous sub-trajectory's initial state.

Next, we define the data likelihood, which models the observed trajectory data $\mathbf{F}$ given the latent states $\boldsymbol{\alpha}_{\mathbf{F}}^{1:B}$:

$$p_\theta(\mathbf{F} | \boldsymbol{\alpha}_{\mathbf{F}}^{1:B}) = \prod_{b=1}^{B} p_\theta(\mathbf{F}^b | \boldsymbol{\alpha}_{\mathbf{F}}^b) = \prod_{b=1}^{B} \prod_{s \in \mathcal{S}^b} p_\theta(\mathbf{F}_s^b | \boldsymbol{\alpha}_{\mathbf{F}}^b) = \prod_{b=1}^{B} \prod_{s \in \mathcal{S}^b} \mathcal{N}(\mathbf{F}_s^b | \hat{\mathbf{F}}_s^b, \sigma_d^2 I), \tag{20}$$

where each sub-trajectory $\mathbf{F}^b$ is modeled as a product of likelihoods over the observed set $\mathcal{S}^b$ and the conditional likelihood for each observed value $\mathbf{F}_s^b$ is Gaussian. Here, $\sigma_d$ controls the observation noise, $\mathcal{S}^b$ denotes observed indices in sub-trajectory $b$, and predictions $\hat{\mathbf{F}}_s^b$ are obtained by applying an MLP to the PDE-solved latent states.

Given parameters $\theta$, the latent states $\boldsymbol{\alpha}_{\mathbf{F}}^{1:B}$ for each trajectory $\mathbf{F}$ are estimated by Maximum-A-Posteriori (MAP):

$$\hat{\boldsymbol{\alpha}}_{\mathbf{F}}^{1:B} = \arg\max_{\boldsymbol{\alpha}_{\mathbf{F}}^{1:B}} p_\theta(\boldsymbol{\alpha}_{\mathbf{F}}^{1:B} | \mathbf{F}) = \arg\max_{\boldsymbol{\alpha}_{\mathbf{F}}^{1:B}} \log p_\theta(\boldsymbol{\alpha}_{\mathbf{F}}^{1:B} | \mathbf{F}). \tag{21}$$

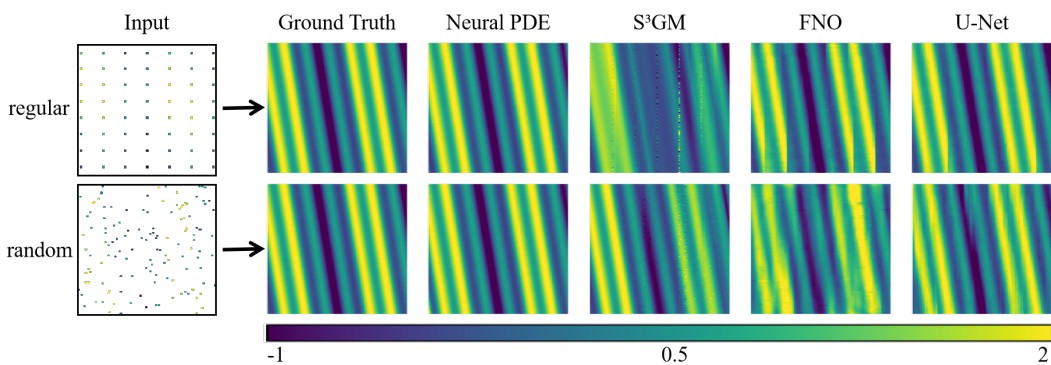

Figure 4: Reconstruction performance for different models on the regular downsampling task with a 16 downsampling factor and the random downsampling task with a 1% downsampling rate.

The parameters $\theta$ are then estimated by maximizing the posterior over all trajectories in the dataset $\mathcal{D}$:

$$\hat{\theta} = \arg\max_{\theta} \sum_{\mathbf{F} \in \mathcal{D}} \max_{\boldsymbol{\alpha}_{\mathbf{F}}^{1:B}} \log p_{\theta}(\boldsymbol{\alpha}_{\mathbf{F}}^{1:B}|\mathbf{F}), \tag{22}$$

which, using Bayes' Law, can be rewritten as:

$$\hat{\theta} = \arg\max_{\theta} \sum_{\mathbf{F} \in \mathcal{D}} \max_{\boldsymbol{\alpha}_{\mathbf{F}}^{1:B}} (\log p_{\theta}(\mathbf{F}|\boldsymbol{\alpha}_{\mathbf{F}}^{1:B}) + \log p_{\theta}(\boldsymbol{\alpha}_{\mathbf{F}}^{1:B})). \tag{23}$$

During training, $\theta$ and $\boldsymbol{\alpha}_{\mathbf{F}}^{1:B}$ are jointly optimized on non-sparse data. The objective, based on Equation 23, is given by:

$$\hat{\theta}, \hat{\boldsymbol{\alpha}}_{\mathbf{F}}^{1:B} = \arg\max_{\theta, \{\boldsymbol{\alpha}_{\mathbf{F}}^{1:B}\}_{\mathbf{F} \in \mathcal{D}}} \sum_{\mathbf{F} \in \mathcal{D}} (\sum_{b=1}^{B} \sum_{s \in \mathcal{S}^b} \log \mathcal{N}(\mathbf{F}_s^b|\hat{\mathbf{F}}_s^b, \sigma_d^2 I)$$
$$+ \log \mathcal{N}(\boldsymbol{\alpha}_{\mathbf{F}}^1|0, \sigma_i^2 I) + \sum_{b=2}^{B} \log \mathcal{N}(\boldsymbol{\alpha}_{\mathbf{F}}^b|\mathbf{H}_{idx}^{b-1}, \sigma_c^2 I)), \tag{24}$$

which is equivalent to minimizing the loss function:

$$\hat{\theta}, \hat{\boldsymbol{\alpha}}_{\mathbf{F}}^{1:B} = \arg\min_{\theta, \{\boldsymbol{\alpha}_{\mathbf{F}}^{1:B}\}_{\mathbf{F} \in \mathcal{D}}} \sum_{\mathbf{F} \in \mathcal{D}} (\sum_{b=1}^{B} \sum_{s \in \mathcal{S}^b} \frac{1}{\sigma_{\text{data}}^2} \mathcal{L}(\mathbf{F}_s^b, \hat{\mathbf{F}}_s^b) + \frac{1}{\sigma_0^2} \mathcal{L}(\boldsymbol{\alpha}_{\mathbf{F}}^1, 0) + \sum_{b=2}^{B} \frac{1}{\sigma_c^2} \mathcal{L}(\boldsymbol{\alpha}_{\mathbf{F}}^b, \mathbf{H}_{idx}^{b-1})), \tag{25}$$

where $\mathcal{L}(x, y)$ represents the L2 loss function.

During testing, the initial latent states $\boldsymbol{\alpha}_{\mathbf{F}}^{1:B}$ for a given trajectory $\mathbf{F}$ are estimated on sparse observations by minimizing the following loss, as derived from Equation 21:

$$\hat{\boldsymbol{\alpha}}_{\mathbf{F}}^{1:B} = \arg\min_{\boldsymbol{\alpha}_{\mathbf{F}}^{1:B}} \sum_{b=1}^{B} \sum_{s \in \mathcal{S}^b} \frac{1}{\sigma_d^2} \mathcal{L}(\mathbf{F}_s^b, \hat{\mathbf{F}}_s^b) + \frac{1}{\sigma_i^2} \mathcal{L}(\boldsymbol{\alpha}_{\mathbf{F}}^1, 0) + \sum_{b=2}^{B} \frac{1}{\sigma_c^2} \mathcal{L}(\boldsymbol{\alpha}_{\mathbf{F}}^b, \mathbf{H}_{idx}^{b-1}). \tag{26}$$

## 4 EXPERIMENTS

This section presents experimental setup and performance of Neural PDE. See Appendix D for details. Code is available at https://anonymous.4open.science/r/Neural-PDE-6AF4.

### 4.1 EXPERIMENTAL SETUP

**Datasets.** Advection Equation and Burgers Equation from PDEBench (Takamoto et al., 2022).

Table 1: Performance comparison of different models on different reconstruction tasks using RMSE (multiplied by 10 for readability). The best results are in bold, and the second-best are underlined.

| Model | Advection | | | | | | Burgers | | | | | |
|---|---|---|---|---|---|---|---|---|---|---|---|---|
| | regular | | | random | | | regular | | | random | | |
| | 4× | 8× | 16× | 5% | 2% | 1% | 4× | 8× | 16× | 5% | 2% | 1% |
| U-Net | 0.214 | 0.297 | 0.517 | 0.463 | 0.662 | 1.054 | 0.151 | **0.254** | 0.910 | 0.442 | 0.550 | 0.948 |
| S³GM | 0.106 | 1.892 | 5.994 | 0.101 | 0.714 | 2.535 | **0.105** | 0.510 | 2.289 | **0.159** | 0.463 | 0.952 |
| FNO | 0.339 | 0.383 | 1.011 | 0.971 | 1.496 | 2.012 | 0.285 | 0.495 | 5.005 | 0.771 | 0.985 | 1.137 |
| LNO | 2.642 | 3.115 | 4.012 | 2.172 | 3.316 | 4.200 | 1.115 | 1.182 | 1.225 | 1.375 | 1.502 | 1.642 |
| DeepONet | 0.229 | 0.268 | 1.054 | 1.221 | 1.354 | 1.698 | 0.425 | 0.779 | 0.920 | 1.105 | 1.269 | 1.513 |
| PINN | 2.456 | 2.551 | 3.555 | 2.422 | 2.712 | 3.162 | 0.246 | 0.547 | 1.321 | 0.256 | 0.473 | 0.968 |
| MMGNet | 0.718 | 2.180 | 4.256 | 1.791 | 4.175 | 4.997 | 0.298 | 0.510 | 0.906 | 1.183 | 2.015 | 2.509 |
| Neural PDE | **0.105** | **0.179** | **0.311** | **0.097** | **0.144** | **0.271** | 0.210 | 0.416 | **0.825** | 0.245 | **0.400** | **0.586** |

Table 2: Comparison of the generalization capability across different tasks.

| Model | Advection | | | | | | Burgers | | | | | |
|---|---|---|---|---|---|---|---|---|---|---|---|---|
| | regular | | | random | | | regular | | | random | | |
| | 4× | 8× | 16× | 5% | 2% | 1% | 4× | 8× | 16× | 5% | 2% | 1% |
| U-Net | 0.295 | 0.711 | 3.034 | 0.552 | 0.662 | 1.325 | 0.698 | 0.625 | 1.448 | 0.520 | 0.550 | 1.074 |
| S³GM | 0.106 | 1.892 | 5.994 | 0.101 | 0.714 | 2.535 | **0.105** | 0.510 | 2.289 | **0.159** | 0.463 | 0.952 |
| FNO | 3.688 | 1.384 | 4.470 | 2.237 | 1.496 | 2.626 | 8.584 | 1.703 | 3.013 | 1.908 | 0.985 | 1.664 |
| LNO | 8.169 | 3.372 | 5.894 | 6.743 | 3.316 | 4.513 | 6.665 | 1.754 | 4.057 | 5.373 | 1.502 | 2.759 |
| DeepONet | 1.540 | 1.292 | 4.201 | 2.311 | 1.354 | 2.415 | 7.020 | 1.462 | 3.956 | 5.401 | 1.269 | 2.644 |
| PINN | 2.456 | 2.551 | 3.555 | 2.422 | 2.712 | 3.162 | 0.246 | 0.547 | 1.321 | 0.256 | 0.473 | 0.968 |
| MMGNet | 0.738 | 2.198 | 4.235 | 1.811 | 4.175 | 5.055 | 0.355 | 0.524 | 0.937 | 1.044 | 2.015 | 2.486 |
| Neural PDE | **0.105** | **0.179** | **0.311** | **0.097** | **0.144** | **0.271** | 0.210 | **0.416** | **0.825** | 0.245 | **0.400** | **0.586** |

**Baselines.** Seven baselines are used to evaluate the effectiveness of our method: two discretized models (U-Net (Ronneberger et al., 2015), S³GM (Li et al., 2024)), three operator-based models (FNO (Li et al., 2021), LNO (Cao et al., 2024), DeepONet (Lu et al., 2021)), and two coordinate-based models, also known as INRs (PINN (Raissi et al., 2019), MMGNet (Luo et al., 2024)).

**Tasks.** We evaluate six reconstruction settings that recover the full spatiotemporal data from sparse observations. These sparse inputs are generated by downsampling the full data using two strategies: regular downsampling, with factors of 4, 8, and 16; and random downsampling, with sampling rates of 0.05, 0.02, and 0.01. Regular downsampling with factor 4 means sampling one point every four points. Random downsampling with rate 0.05 means randomly choosing 5% of the points. We focus exclusively on reconstruction tasks rather than prediction tasks, as reconstruction focuses on recovering missing data across both spatial and temporal dimensions, whereas prediction focuses only on modeling evolution over temporal dimension.

**Implementation.** Neural PDE is implemented by PyTorch and torchdiffeq. All experiments are conducted on NVIDIA 4090. The neural networks within Neural PDE utilize MLPs with a hidden layer dimension of 64 and tanh as the activation function. The MLP responsible for mapping the latent state to the final output employs ReLU as the activation function. The latent state dimension is 32. Sub-trajectories have a length of 9 with an overlap of 2. We set $1/\sigma_d^2$ to 1, and select $1/\sigma_i^2$ and $1/\sigma_c^2$ from $\{0, 0.5, 1\}$. During training and testing, we optimize the model with the Adam optimizer for 1,000 epochs at a learning rate of 0.01.

## 4.2 EXPERIMENTAL RESULTS

**Reconstruction Performance.** To evaluate reconstruction performance, the models are trained on different tasks and tested on the same tasks used during training. Table 1 shows the performance of various models. As the sparsity of the data increases, the reconstruction performance of all models declines. However, our proposed model demonstrates remarkable stability, particularly under extreme sparsity conditions. Specifically, for 16× regular and 1% random downsampling, it reduces RMSE by 39.85% and 74.29% on advection equation, and by 8.94% and 38.18% on Burgers

Table 3: The result of continuity evaluation experiment.

| Model | Advection | | | | | | Burgers | | | | | |
|---|---|---|---|---|---|---|---|---|---|---|---|---|
| | regular | | | random | | | regular | | | random | | |
| | 4× | 8× | 16× | 5% | 2% | 1% | 4× | 8× | 16× | 5% | 2% | 1% |
| U-Net | 0.432 | 1.248 | 2.753 | 0.699 | 1.755 | 1.829 | 0.202 | 0.386 | 1.001 | 0.411 | 0.522 | 0.853 |
| S³GM | 0.965 | 2.957 | 6.116 | 0.579 | 1.579 | 2.886 | 0.338 | 1.592 | 3.934 | 0.292 | 0.627 | 0.993 |
| FNO | 0.494 | 4.510 | 8.361 | 0.827 | 1.324 | 1.714 | 0.358 | 0.640 | 5.005 | 0.662 | 0.853 | 1.019 |
| LNO | 2.644 | 3.121 | 4.006 | 1.878 | 3.171 | 4.063 | 1.116 | 1.181 | 1.219 | 1.324 | 1.418 | 1.513 |
| DeepONet | 0.933 | 0.947 | 4.308 | 1.233 | 1.229 | 1.529 | 0.518 | 0.796 | 3.305 | 1.026 | 1.164 | 1.357 |
| PINN | 2.441 | 2.518 | 3.368 | 2.381 | 2.459 | 2.686 | 0.233 | 0.528 | 1.162 | 0.243 | 0.457 | 0.943 |
| MMGNet | 0.716 | 2.170 | 4.162 | 0.499 | 2.466 | 4.091 | 0.297 | 0.504 | 0.902 | 0.560 | 1.301 | 2.053 |
| Neural PDE | **0.122** | **0.193** | **0.270** | **0.100** | **0.119** | **0.161** | **0.188** | **0.359** | **0.752** | **0.192** | **0.274** | **0.391** |

equation, compared to the second-best method. Figure 4 shows only our model accurately reconstructs the spatiotemporal dynamics. These results highlight the model's exceptional capability to reconstruct complex spatiotemporal dynamics even from extremely sparse data.

**Generalization Capability.** To evaluate generalization capability across different tasks, the models are trained on the 2% random downsampling task and tested on all six tasks, as shown in Figure 5. As shown in Table 2, for 16× regular and 1% random downsampling, our model reduces RMSE by 89.74% and 79.54% on advection equation, and by 11.95% and 38.44% on Burgers equation, compared to the second-best method. Comparing Table 1 and Table 2, the pre-trained models (Neural PDE and S³GM) employ a task-agnostic training approach, leading to consistent performance. In contrast, end-to-end models suffer significant performance degradation when tested on unseen tasks. This highlights the superior generalization of pre-trained models.

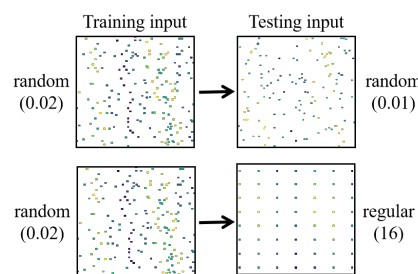

Figure 5: Training and testing tasks for generalization capability evaluation.

**Continuity Evaluation.** To evaluate the continuity, the models are tested on tasks with a resolution twice that of the training tasks, as shown in Figure 6. The spatial dimension of the testing output (256) is twice that of the training output (128), while the temporal dimension remains fixed at 100. As shown in Table 3, for 16× regular and 1% random downsampling, our model reduces RMSE by 90.19% and 89.47% on advection equation, and by 16.62% and 54.16% on Burgers equation, compared to the second-best method. Comparing Table 1 and Table 3, discretized models (U-Net and S³GM) degrade at higher resolution because smaller grid cells shrink their receptive fields. Neural operator-based models (FNO, LNO, and DeepONet) improve on random downsampling tasks but drop on regular downsampling tasks. For Neural PDE, higher resolution enables a smaller integration step size, yielding improvements on most tasks and demonstrating its continuous modeling advantage.

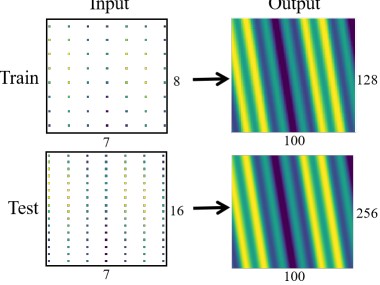

Figure 6: Training and testing tasks for continuity evaluation. Both the input and output resolutions of the test tasks are twice those of the training tasks.

## 5 CONCLUSION

We propose Neural PDE to model spatiotemporal dynamics continuously. It incorporates a generative pretraining framework with an auto-decoder to optimize initial latent. To enable efficient training, we introduce an overlapping multiple shooting technique and ensures smooth transitions using a continuity loss. Experiments show that Neural PDE achieves state-of-the-art performance, demonstrating strong continuum modeling and generalization capability.

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

## A    LLM USAGE STATEMENT

We used LLM for grammar checking and language polishing to improve readability.

## B    PROOFS

### B.1    THE CASE OF BINARY FUNCTIONS

We employ Neural ODEs to model binary functions $f(x, y)$:

$$\frac{\partial f}{\partial x}(x, y) = \mathcal{F}_1\left(x, f(x, y); \theta_1\right), \tag{27}$$

$$\frac{\partial f}{\partial y}(x, y) = \mathcal{F}_2\left(y, f(x, y); \theta_2\right). \tag{28}$$

**Proposition B.1.** *The vector field $(\mathcal{F}_1, \mathcal{F}_2)$ is non-conservative, i.e., it does not satisfy path independence.*

*Proof.* Given an initial condition $f(x_0, y_0)$, we examine two possible integration paths to solve for $f(x, y)$. First, we integrate with respect to $x$ and then with respect to $y$:

$$\begin{aligned} f(x, y) &= f(x_0, y_0) + \int_{x_0}^{x} \frac{\partial f}{\partial x}(x', y_0)dx' + \int_{y_0}^{y} \frac{\partial f}{\partial y}(x, y')dy' \\ &= f(x_0, y_0) + \int_{x_0}^{x} \mathcal{F}_1\left(x', f(x', y_0); \theta_1\right) dx' + \int_{y_0}^{y} \mathcal{F}_2\left(y', f(x, y'); \theta_2\right) dy'. \end{aligned} \tag{29}$$

Next, we integrate with respect to $y$ and then $x$:

$$\begin{aligned} f(x, y) &= f\left(x_0, y_0\right) + \int_{y_0}^{y} \frac{\partial f}{\partial y}\left(x_0, y'\right) dy' + \int_{x_0}^{x} \frac{\partial f}{\partial x}\left(x', y\right) dx' \\ &= f\left(x_0, y_0\right) + \int_{x_0}^{x} \frac{\partial f}{\partial x}\left(x', y\right) dx' + \int_{y_0}^{y} \frac{\partial f}{\partial y}\left(x_0, y'\right) dy' \\ &= f\left(x_0, y_0\right) + \int_{x_0}^{x} \mathcal{F}_1\left(x', f(x', y); \theta_1\right) dx' + \int_{y_0}^{y} \mathcal{F}_2\left(y', f(x_0, y'); \theta_2\right) dy'. \end{aligned} \tag{30}$$

Comparing the results obtained from the two different integration paths, we observe that:

$$\begin{aligned} \mathcal{F}_1\left(x', f(x', y_0); \theta_1\right) &\neq \mathcal{F}_1\left(x', f(x', y); \theta_1\right) \\ \mathcal{F}_2\left(y', f(x, y'); \theta_2\right) &\neq \mathcal{F}_2\left(y', f(x_0, y'); \theta_2\right) \end{aligned} \tag{31}$$

Furthermore, since the neural networks $\mathcal{F}_1$ and $\mathcal{F}_2$ are independent of each other, the results obtained from the two different integration paths are not equivalent, and path independence is not satisfied. □

We employ Neural PDE to model binary functions $f(x, y)$:

$$\begin{aligned} \frac{\partial f}{\partial x}(x, y) &= \frac{\partial f}{\partial x}(x, y_0) + \int_{y_0}^{y} \frac{\partial^2 f}{\partial x \partial y}(x, y')dy' \\ &= \mathcal{F}_x(x, y_0, f(x, y_0); \theta_x) + \int_{y_0}^{y} \mathcal{F}(x, y', f(x, y'); \theta)dy', \end{aligned} \tag{32}$$

$$\begin{aligned} \frac{\partial f}{\partial y}(x, y) &= \frac{\partial f}{\partial y}(x_0, y) + \int_{x_0}^{x} \frac{\partial^2 f}{\partial y \partial x}(x', y) dx' \\ &= \mathcal{F}_y(x_0, y, f(x_0, y); \theta_y) + \int_{x_0}^{x} \mathcal{F}(x', y, f(x', y); \theta)dx'. \end{aligned} \tag{33}$$

**Proposition B.2.** *The vector field formed by the expressions on the right sides of Equation 32 and Equation 33 is conservative, i.e., it satisfies path independence. The integration result is given by:*

$$f(x, y) = f(x_0, y_0) + \int_{x_0}^{x} \mathcal{F}_x(x', y_0, f(x', y_0); \theta_x) dx' + \int_{y_0}^{y} \mathcal{F}_y(x_0, y', f(x_0, y'); \theta_y) dy'$$

$$+ \int_{y_0}^{y} \int_{x_0}^{x} \mathcal{F}(x', y', f(x', y'); \theta) dx' dy'. \tag{34}$$

*Proof.* For the vector field $(P, Q)$, where:

$$P(x, y) = \mathcal{F}_x(x, y_0, f(x, y_0); \theta_x) + \int_{y_0}^{y} \mathcal{F}(x, y', f(x, y'); \theta) dy', \tag{35}$$

$$Q(x, y) = \mathcal{F}_y(x_0, y, f(x_0, y); \theta_y) + \int_{x_0}^{x} \mathcal{F}(x', y, f(x', y); \theta) dx'. \tag{36}$$

we compute the following partial derivatives:

$$\frac{\partial P}{\partial y}(x, y) = \mathcal{F}(x, y, f(x, y); \theta), \tag{37}$$

$$\frac{\partial Q}{\partial x}(x, y) = \mathcal{F}(x, y, f(x, y); \theta). \tag{38}$$

Thus, we obtain:

$$\frac{\partial Q}{\partial x}(x, y) - \frac{\partial P}{\partial y}(x, y) = 0. \tag{39}$$

Since the curl of the vector field $(P, Q)$ is zero, the vector field is conservative and satisfies path independence.

Given an initial condition $f(x_0, y_0)$, we examine two possible integration paths to solve for $f(x, y)$. First, we integrate with respect to $x$ and then with respect to $y$:

$$f(x, y) = f(x_0, y_0) + \int_{x_0}^{x} \frac{\partial f}{\partial x}(x', y_0) dx' + \int_{y_0}^{y} \frac{\partial f}{\partial y}(x, y') dy'$$

$$= f(x_0, y_0) + \int_{x_0}^{x} \frac{\partial f}{\partial x}(x', y_0) dx' + \int_{y_0}^{y} \frac{\partial f}{\partial y}(x_0, y') + \int_{x_0}^{x} \frac{\partial^2 f}{\partial x \partial y}(x', y') dx' dy'$$

$$= f(x_0, y_0) + \int_{x_0}^{x} \frac{\partial f}{\partial x}(x', y_0) dx' + \int_{y_0}^{y} \frac{\partial f}{\partial y}(x_0, y') dy' + \int_{y_0}^{y} \int_{x_0}^{x} \frac{\partial^2 f}{\partial x \partial y}(x', y') dx' dy'$$

$$= f(x_0, y_0) + \int_{x_0}^{x} \mathcal{F}_x(x', y_0, f(x', y_0); \theta_x) dx' + \int_{y_0}^{y} \mathcal{F}_y(x_0, y', f(x_0, y'); \theta_y) dy'$$

$$+ \int_{y_0}^{y} \int_{x_0}^{x} \mathcal{F}(x', y', f(x', y'); \theta) dx' dy'. \tag{40}$$

Next, we integrate with respect to $y$ and then $x$:

$$f(x, y) = f(x_0, y_0) + \int_{y_0}^{y} \frac{\partial f}{\partial y}(x_0, y') dy' + \int_{x_0}^{x} \frac{\partial f}{\partial x}(x', y) dx'$$

$$= f(x_0, y_0) + \int_{y_0}^{y} \frac{\partial f}{\partial y}(x_0, y') dy' + \int_{x_0}^{x} \frac{\partial f}{\partial x}(x', y_0) + \int_{y_0}^{y} \frac{\partial^2 f}{\partial y \partial x}(x', y') dy' dx'$$

$$= f(x_0, y_0) + \int_{y_0}^{y} \frac{\partial f}{\partial y}(x_0, y') dy' + \int_{x_0}^{x} \frac{\partial f}{\partial x}(x', y_0) dx' + \int_{x_0}^{x} \int_{y_0}^{y} \frac{\partial^2 f}{\partial y \partial x}(x', y') dy' dx'$$

$$= f(x_0, y_0) + \int_{x_0}^{x} \frac{\partial f}{\partial x}(x', y_0) dx' + \int_{y_0}^{y} \frac{\partial f}{\partial y}(x_0, y') dy' + \int_{y_0}^{y} \int_{x_0}^{x} \frac{\partial^2 f}{\partial x \partial y}(x', y') dx' dy'$$

$$= f(x_0, y_0) + \int_{x_0}^{x} \mathcal{F}_x(x', y_0, f(x', y_0); \theta_x) dx' + \int_{y_0}^{y} \mathcal{F}_y(x_0, y', f(x_0, y'); \theta_y) dy'$$

$$+ \int_{y_0}^{y} \int_{x_0}^{x} \mathcal{F}(x', y', f(x', y'); \theta) dx' dy'. \tag{41}$$

We can conclude that the results obtained from the two different integration paths are the same, given by:

$$
\begin{aligned}
f(x,y) =\,& f(x_0, y_0) + \int_{x_0}^{x} \mathcal{F}_x(x', y_0, f(x', y_0); \theta_x) dx' + \int_{y_0}^{y} \mathcal{F}_y(x_0, y', f(x_0, y'); \theta_y) dy' \\
& + \int_{y_0}^{y} \int_{x_0}^{x} \mathcal{F}(x', y', f(x', y'); \theta) dx' dy'.
\end{aligned}
\tag{42}
$$

□

**Proposition B.3.** *Neural PDE can be solved using nested ODE solvers, with the solution process encapsulated in the function PDESolve:*

$$
f(x, y) = PDESolve(f(x_0, y_0), x_0, x, y_0, y).
\tag{43}
$$

*Proof.* We begin by expressing $f(x, y)$ as follows:

$$
f(x, y) = f(x_0, y_0) + \int_{x_0}^{x} \frac{\partial f}{\partial x}(x', y_0) dx' + \int_{y_0}^{y} \frac{\partial f}{\partial y}(x, y') dy'.
\tag{44}
$$

This expression can be interpreted as solving a pair of nested ODEs:

$$
f(x, y) = \text{ODESolve}(\text{ODESolve}(f(x_0, y_0), \frac{\partial f}{\partial x}(x', y_0), x_0, x), \frac{\partial f}{\partial y}(x, y'), y_0, y).
\tag{45}
$$

Next, we consider the specific forms of the partial derivatives. From the PDE governing the derivative with respect to $x$ (Equation 7), we have:

$$
\frac{\partial f}{\partial x}(x', y_0) = \mathcal{F}_x(x', y_0, f(x', y_0); \theta_x),
\tag{46}
$$

where $\mathcal{F}_x(x', y_0, f(x', y_0); \theta_x)$ is a neural network that parametrizes the derivative of $f(x, y)$ with respect to $x$ at $y = y_0$.

Similarly, from the PDE governing the derivative with respect to $y$ (Equation 8), we have:

$$
\frac{\partial f}{\partial y}(x, y') = \frac{\partial f}{\partial y}(x_0, y') + \int_{x_0}^{x} \frac{\partial^2 f}{\partial x \partial y}(x', y') dx',
\tag{47}
$$

which can also be solved using an ODE solver:

$$
\frac{\partial f}{\partial y}(x, y') = \text{ODESolve}(\frac{\partial f}{\partial y}(x_0, y'), \frac{\partial^2 f}{\partial x \partial y}(x', y'), x_0, x).
\tag{48}
$$

Here, the initial condition $\frac{\partial f}{\partial y}(x_0, y')$ is given by:

$$
\frac{\partial f}{\partial y}(x_0, y') = \mathcal{F}_y(x_0, y', f(x_0, y'); \theta_y),
\tag{49}
$$

and the mixed partial derivative is expressed as:

$$
\frac{\partial^2 f}{\partial x \partial y}(x', y') = \mathcal{F}(x', y', f(x', y'); \theta).
\tag{50}
$$

We have shown that the solution to the neural PDE can be reduced to the solution of nested ODEs, where each ODE is governed by a neural network that parametrizes the required derivatives. This process is encapsulated by the function PDESolve. Therefore, the solution to the Neural PDE can be expressed as:

$$
f(x, y) = \text{PDESolve}(f(x_0, y_0), x_0, x, y_0, y).
\tag{51}
$$

□

## B.2 THE CASE OF TERNARY FUNCTIONS

We employ Neural ODEs to model ternary functions $f(x, y, z)$:

$$\frac{\partial f}{\partial x}(x, y, z) = \mathcal{F}_1 \left( x, f(x, y, z); \theta_1 \right), \tag{52}$$

$$\frac{\partial f}{\partial y}(x, y, z) = \mathcal{F}_2 \left( y, f(x, y, z); \theta_2 \right), \tag{53}$$

$$\frac{\partial f}{\partial z}(x, y, z) = \mathcal{F}_3 \left( z, f(x, y, z); \theta_3 \right). \tag{54}$$

**Proposition B.4.** *The vector field $(\mathcal{F}_1, \mathcal{F}_2, \mathcal{F}_3)$ is non-conservative, i.e., it does not satisfy path independence.*

*Proof.* Given an initial condition $f(x_0, y_0, z_0)$, we examine two possible integration paths to solve for $f(x, y, z)$. First, we integrate with respect to $x$, then with respect to $y$, and with respect to $z$:

$$
\begin{aligned}
f(x, y, z) =& f(x_0, y_0, z_0) + \int_{x_0}^{x} \frac{\partial f}{\partial x}(x', y_0, z_0) dx' + \int_{y_0}^{y} \frac{\partial f}{\partial y}(x, y', z_0) dy' + \int_{z_0}^{z} \frac{\partial f}{\partial z}(x, y, z') dz' \\
=& f(x_0, y_0, z_0) + \int_{x_0}^{x} \mathcal{F}_1 \left( x', f(x', y_0, z_0); \theta_1 \right) dx' \\
&+ \int_{y_0}^{y} \mathcal{F}_2 \left( y', f(x, y', z_0); \theta_2 \right) dy' + \int_{z_0}^{z} \mathcal{F}_3 \left( z', f(x, y, z'); \theta_3 \right) dz'.
\end{aligned}
\tag{55}
$$

Next, we integrate with respect to $z$, then $y$, and finally $x$:

$$
\begin{aligned}
f(x, y, z) =& f \left( x_0, y_0, z_0 \right) + \int_{z_0}^{z} \frac{\partial f}{\partial z} \left( x_0, y_0, z' \right) dz' + \int_{y_0}^{y} \frac{\partial f}{\partial y} \left( x_0, y', z \right) dy' + \int_{x_0}^{x} \frac{\partial f}{\partial x} \left( x', y, z \right) dx' \\
=& f \left( x_0, y_0, z_0 \right) + \int_{x_0}^{x} \frac{\partial f}{\partial x} \left( x', y, z \right) dx' + \int_{y_0}^{y} \frac{\partial f}{\partial y} \left( x_0, y', z \right) dy' + \int_{z_0}^{z} \frac{\partial f}{\partial z} \left( x_0, y_0, z' \right) dz' \\
=& f \left( x_0, y_0, z_0 \right) + \int_{x_0}^{x} \mathcal{F}_1 \left( x', f(x', y, z); \theta_1 \right) dx' \\
&+ \int_{y_0}^{y} \mathcal{F}_2 \left( y', f(x_0, y', z); \theta_2 \right) dy' + \int_{z_0}^{z} \mathcal{F}_3 \left( z', f(x_0, y_0, z'); \theta_3 \right) dz'.
\end{aligned}
\tag{56}
$$

Comparing the results obtained from the two different integration paths, we observe that:

$$
\begin{aligned}
\mathcal{F}_1 \left( x', f(x', y_0, z_0); \theta_1 \right) &\neq \mathcal{F}_1 \left( x', f(x', y, z); \theta_1 \right) \\
\mathcal{F}_2 \left( y', f(x, y', z_0); \theta_2 \right) &\neq \mathcal{F}_2 \left( y', f(x_0, y', z); \theta_2 \right) \\
\mathcal{F}_3 \left( z', f(x, y, z'); \theta_3 \right) &\neq \mathcal{F}_3 \left( z', f(x_0, y_0, z'); \theta_3 \right)
\end{aligned}
\tag{57}
$$

Furthermore, since the neural networks $\mathcal{F}_1$, $\mathcal{F}_2$ and $\mathcal{F}_3$ are independent of each other, the results obtained from the two different integration paths are not equivalent, and path independence is not satisfied. $\qquad\square$

We employ Neural PDE to model ternary functions $f(x, y, z)$:

$$\begin{aligned}
\frac{\partial f}{\partial x}(x, y, z) =& \frac{\partial f}{\partial x}(x, y_0, z_0) + \int_{y_0}^{y} \frac{\partial^2 f}{\partial y \partial x}(x, y', z_0) dy' + \int_{z_0}^{z} \frac{\partial^2 f}{\partial z \partial x}(x, y, z') dz' \\
=& \frac{\partial f}{\partial x}(x, y_0, z_0) + \int_{y_0}^{y} \frac{\partial^2 f}{\partial y \partial x}(x, y', z_0) dy' \\
& + \int_{z_0}^{z} \frac{\partial^2 f}{\partial z \partial x}(x, y_0, z') + \int_{y_0}^{y} \frac{\partial^3 f}{\partial y \partial z \partial x}(x, y', z') dy' dz' \\
=& \mathcal{F}_x(x, y_0, z_0, f(x, y_0, z_0); \theta_x) + \int_{y_0}^{y} \mathcal{F}_{xy}(x, y', z_0, f(x, y', z_0); \theta_{xy}) dy' \\
& + \int_{z_0}^{z} \mathcal{F}_{xz}(x, y_0, z', f(x, y_0, z'); \theta_{xz}) + \int_{y_0}^{y} \mathcal{F}(x, y', z', f(x, y', z'); \theta) dy' dz',
\end{aligned}$$
$$(58)$$

$$\begin{aligned}
\frac{\partial f}{\partial y}(x, y, z) =& \frac{\partial f}{\partial y}(x_0, y, z_0) + \int_{x_0}^{x} \frac{\partial^2 f}{\partial x \partial y}(x', y, z_0) dx' + \int_{z_0}^{z} \frac{\partial^2 f}{\partial z \partial y}(x, y, z') dz' \\
=& \frac{\partial f}{\partial y}(x_0, y, z_0) + \int_{x_0}^{x} \frac{\partial^2 f}{\partial x \partial y}(x', y, z_0) dx' \\
& + \int_{z_0}^{z} \frac{\partial^2 f}{\partial z \partial y}(x_0, y, z') + \int_{x_0}^{x} \frac{\partial^3 f}{\partial x \partial z \partial y}(x', y, z') dx' dz' \\
=& \mathcal{F}_y(x_0, y, z_0, f(x_0, y, z_0); \theta_y) + \int_{x_0}^{x} \mathcal{F}_{xy}(x', y, z_0, f(x', y, z_0); \theta_{xy}) dx' \\
& + \int_{z_0}^{z} \mathcal{F}_{yz}(x_0, y, z', f(x_0, y, z'); \theta_{yz}) + \int_{x_0}^{x} \mathcal{F}(x', y, z', f(x', y, z'), \theta) dx' dz',
\end{aligned}$$
$$(59)$$

$$\begin{aligned}
\frac{\partial f}{\partial z}(x, y, z) =& \frac{\partial f}{\partial z}(x_0, y_0, z) + \int_{x_0}^{x} \frac{\partial^2 f}{\partial x \partial z}(x', y_0, z) dx' + \int_{y_0}^{y} \frac{\partial^2 f}{\partial y \partial z}(x, y', z) dy' \\
=& \frac{\partial f}{\partial z}(x_0, y_0, z) + \int_{x_0}^{x} \frac{\partial^2 f}{\partial x \partial z}(x', y_0, z) dx' \\
& + \int_{y_0}^{y} \frac{\partial^2 f}{\partial y \partial z}(x_0, y', z) + \int_{x_0}^{x} \frac{\partial^3 f}{\partial x \partial y \partial z}(x', y', z) dx' dy' \\
=& \mathcal{F}_z(x_0, y_0, z, f(x_0, y_0, z); \theta_z) + \int_{x_0}^{x} \mathcal{F}_{xz}(x', y_0, z, f(x', y_0, z); \theta_{xz}) dx' \\
& + \int_{y_0}^{y} \mathcal{F}_{yz}(x_0, y', z, f(x_0, y', z); \theta_{yz}) + \int_{x_0}^{x} \mathcal{F}(x', y', z, f(x', y', z), \theta) dx' dy'.
\end{aligned}$$
$$(60)$$

**Proposition B.5.** *The vector field formed by the expressions on the right sides of Equation 58, Equation 59, and Equation 60 is conservative, i.e., it satisfies path independence. The integration*

*result is given by:*

$$
\begin{aligned}
f(x, y, z) =& f(x_0, y_0, z_0) + \int_{x_0}^{x} \mathcal{F}_x(x', y_0, z_0, f(x', y_0, z_0); \theta_x) dx' \\
&+ \int_{y_0}^{y} \mathcal{F}_y(x_0, y', z_0, f(x_0, y', z_0); \theta_y) dy' + \int_{z_0}^{z} \mathcal{F}_z(x_0, y_0, z', f(x_0, y_0, z'); \theta_z) dz' \\
&+ \int_{y_0}^{y} \int_{x_0}^{x} \mathcal{F}_{xy}(x', y', z_0, f(x', y', z_0); \theta_{xy}) dx' dy' \\
&+ \int_{z_0}^{z} \int_{x_0}^{x} \mathcal{F}_{xz}(x', y_0, z', f(x', y_0, z'); \theta_{xz}) dx' dz' \\
&+ \int_{z_0}^{z} \int_{y_0}^{y} \mathcal{F}_{yz}(x_0, y', z', f(x_0, y', z'); \theta_{yz}) dy' dz' \\
&+ \int_{z_0}^{z} \int_{y_0}^{y} \int_{x_0}^{x} \mathcal{F}(x', y', z', f(x', y', z'); \theta) dx' dy' dz'.
\end{aligned}
\tag{61}
$$

*Proof.* For the vector field $(P, Q, R)$, where:

$$
\begin{aligned}
P(x, y, z) =& \mathcal{F}_x(x, y_0, z_0, f(x, y_0, z_0); \theta_x) + \int_{y_0}^{y} \mathcal{F}_{xy}(x, y', z_0, f(x, y', z_0); \theta_{xy}) dy' \\
&+ \int_{z_0}^{z} \mathcal{F}_{xz}(x, y_0, z', f(x, y_0, z'); \theta_{xz}) + \int_{y_0}^{y} \mathcal{F}(x, y', z', f(x, y', z'); \theta) dy' dz',
\end{aligned}
\tag{62}
$$

$$
\begin{aligned}
Q(x, y, z) =& \mathcal{F}_y(x_0, y, z_0, f(x_0, y, z_0); \theta_y) + \int_{x_0}^{x} \mathcal{F}_{xy}(x', y, z_0, f(x', y, z_0); \theta_{xy}) dx' \\
&+ \int_{z_0}^{z} \mathcal{F}_{yz}(x_0, y, z', f(x_0, y, z'); \theta_{yz}) + \int_{x_0}^{x} \mathcal{F}(x', y, z', f(x', y, z'), \theta) dx' dz',
\end{aligned}
\tag{63}
$$

$$
\begin{aligned}
R(x, y, z) =& \mathcal{F}_z(x_0, y_0, z, f(x_0, y_0, z); \theta_z) + \int_{x_0}^{x} \mathcal{F}_{xz}(x', y_0, z, f(x', y_0, z); \theta_{xz}) dx' \\
&+ \int_{y_0}^{y} \mathcal{F}_{yz}(x_0, y', z, f(x_0, y', z); \theta_{yz}) + \int_{x_0}^{x} \mathcal{F}(x', y', z, f(x', y', z), \theta) dx' dy'.
\end{aligned}
\tag{64}
$$

we compute the following partial derivatives:

$$
\frac{\partial P}{\partial y}(x, y, z) = \mathcal{F}_{xy}(x, y, z_0, f(x, y, z_0); \theta_{xy}) + \int_{z_0}^{z} \mathcal{F}(x, y, z', f(x, y, z'); \theta) dz',
\tag{65}
$$

$$
\frac{\partial P}{\partial z}(x, y, z) = \mathcal{F}_{xz}(x, y_0, z, f(x, y_0, z); \theta_{xz}) + \int_{y_0}^{y} \mathcal{F}(x, y', z, f(x, y', z); \theta) dy',
\tag{66}
$$

$$
\frac{\partial Q}{\partial x}(x, y, z) = \mathcal{F}_{xy}(x, y, z_0, f(x, y, z_0); \theta_{xy}) + \int_{z_0}^{z} \mathcal{F}(x, y, z', f(x, y, z'); \theta) dz',
\tag{67}
$$

$$
\frac{\partial Q}{\partial z}(x, y, z) = \mathcal{F}_{yz}(x_0, y, z, f(x_0, y, z); \theta_{yz}) + \int_{x_0}^{x} \mathcal{F}(x', y, z, f(x', y, z), \theta) dx',
\tag{68}
$$

$$
\frac{\partial R}{\partial x}(x, y, z) = \mathcal{F}_{xz}(x, y_0, z, f(x, y_0, z); \theta_{xz}) + \int_{y_0}^{y} \mathcal{F}(x, y', z, f(x, y', z); \theta) dy',
\tag{69}
$$

$$
\frac{\partial R}{\partial y}(x, y, z) = \mathcal{F}_{yz}(x_0, y, z, f(x_0, y, z); \theta_{yz}) + \int_{x_0}^{x} \mathcal{F}(x', y, z, f(x', y, z), \theta) dx'.
\tag{70}
$$

Thus, we obtain:

$$
\frac{\partial R}{\partial y}(x, y, z) - \frac{\partial Q}{\partial z}(x, y, z) = 0,
\tag{71}
$$

$$\frac{\partial p}{\partial z}(x, y, z) - \frac{\partial R}{\partial x}(x, y, z) = 0, \tag{72}$$

$$\frac{\partial Q}{\partial x}(x, y, z) - \frac{\partial P}{\partial y}(x, y, z) = 0. \tag{73}$$

Since the curl of the vector field $(P, Q, R)$ is zero, the vector field is conservative and satisfies path independence.

Given an initial condition $f(x_0, y_0, z_0)$, we integrate with respect to $x$, then with respect to $y$, and finally with respect to $z$:

$$
\begin{aligned}
f(x, y, z) =& f\left(x_0, y_0, z_0\right) + \int_{x_0}^{x} \frac{\partial f}{\partial x}\left(x', y_0, z_0\right) dx' + \int_{y_0}^{y} \frac{\partial f}{\partial y}\left(x, y', z_0\right) dy' + \int_{z_0}^{z} \frac{\partial f}{\partial z}\left(x, y, z'\right) dz' \\
=& f\left(x_0, y_0, z_0\right) + \int_{x_0}^{x} \frac{\partial f}{\partial x}\left(x', y_0, z_0\right) dx' \\
& + \int_{y_0}^{y} \frac{\partial f}{\partial y}(x_0, y', z_0) + \int_{x_0}^{x} \frac{\partial^2 f}{\partial x \partial y}(x', y', z_0) dx' dy' \\
& + \int_{z_0}^{z} \frac{\partial f}{\partial z}\left(x_0, y_0, z'\right) + \int_{x_0}^{x} \frac{\partial^2 f}{\partial x \partial z}(x', y_0, z') dx' \\
& + \int_{y_0}^{y} \frac{\partial^2 f}{\partial y \partial z}(x_0, y', z') + \int_{x_0}^{x} \frac{\partial^3 f}{\partial x \partial y \partial z}(x', y', z') dx' dy' dz' \\
=& f\left(x_0, y_0, z_0\right) + \int_{x_0}^{x} \frac{\partial f}{\partial x}\left(x', y_0, z_0\right) dx' \\
& + \int_{y_0}^{y} \frac{\partial f}{\partial y}(x_0, y', z_0) dy' + \int_{y_0}^{y} \int_{x_0}^{x} \frac{\partial^2 f}{\partial x \partial y}(x', y', z_0) dx' dy' \\
& + \int_{z_0}^{z} \frac{\partial f}{\partial z}\left(x_0, y_0, z'\right) dz' + \int_{z_0}^{z} \int_{x_0}^{x} \frac{\partial^2 f}{\partial x \partial z}(x', y_0, z') dx' dz' \\
& + \int_{z_0}^{z} \int_{y_0}^{y} \frac{\partial^2 f}{\partial y \partial z}(x_0, y', z') dy' dz' + \int_{z_0}^{z} \int_{y_0}^{y} \int_{x_0}^{x} \frac{\partial^3 f}{\partial x \partial y \partial z}(x', y', z') dx' dy' dz' \\
=& f\left(x_0, y_0, z_0\right) + \int_{x_0}^{x} \mathcal{F}_x(x', y_0, z_0, f(x', y_0, z_0); \theta_x) dx' \\
& + \int_{y_0}^{y} \mathcal{F}_y(x_0, y', z_0, f(x_0, y', z_0); \theta_y) dy' + \int_{z_0}^{z} \mathcal{F}_z(x_0, y_0, z', f(x_0, y_0, z'); \theta_z) dz' \\
& + \int_{y_0}^{y} \int_{x_0}^{x} \mathcal{F}_{xy}(x', y', z_0, f(x', y', z_0); \theta_{xy}) dx' dy' \\
& + \int_{z_0}^{z} \int_{x_0}^{x} \mathcal{F}_{xz}(x', y_0, z', f(x', y_0, z'); \theta_{xz}) dx' dz' \\
& + \int_{z_0}^{z} \int_{y_0}^{y} \mathcal{F}_{yz}(x_0, y', z', f(x_0, y', z'); \theta_{yz}) dy' dz' \\
& + \int_{z_0}^{z} \int_{y_0}^{y} \int_{x_0}^{x} \mathcal{F}(x', y', z', f(x', y', z'); \theta) dx' dy' dz'.
\end{aligned}
\tag{74}
$$

$\square$

## C  DERIVATION OF LOSS FUNCTION

To estimate the parameters $\theta$ of the Neural PDE, as well as the initial latent states $\boldsymbol{\alpha}_{\mathbf{F}}^{1:B}$ for each trajectory $\mathbf{F}$, we adopt a probabilistic framework.

We begin by defining the continuity inducing prior as follows:

$$p_\theta(\boldsymbol{\alpha}_{\mathbf{F}}^{1:B}) = p(\boldsymbol{\alpha}_{\mathbf{F}}^1) \prod_{b=2}^B p_\theta(\boldsymbol{\alpha}_{\mathbf{F}}^b | \boldsymbol{\alpha}_{\mathbf{F}}^{b-1}), \tag{75}$$

where the initial state $\boldsymbol{\alpha}_{\mathbf{F}}^1$ follows a zero-mean Gaussian distribution with covariance $\sigma_i^2 I$:

$$p(\boldsymbol{\alpha}_{\mathbf{F}}^1) = \mathcal{N}(\boldsymbol{\alpha}_{\mathbf{F}}^1 | 0, \sigma_i^2 I), \tag{76}$$

and the initial state of each subsequent sub-trajectory is modeled conditionally on the initial state of the preceding sub-trajectory:

$$p_\theta(\boldsymbol{\alpha}_{\mathbf{F}}^b | \boldsymbol{\alpha}_{\mathbf{F}}^{b-1}) = \mathcal{N}(\boldsymbol{\alpha}_{\mathbf{F}}^b | \mathbf{H}_{idx}^{b-1}, \sigma_c^2 I), \tag{77}$$

with $b = 2, \ldots, B$. Here, $\mathcal{N}(\mu, \Sigma)$ denotes a multivariate Gaussian distribution, $I \in \mathbb{R}^{D \times D}$ is the identity matrix, and the parameters $\sigma_i$ and $\sigma_c$ control the regularization strength for the initial state and the continuity, respectively.

The term $\mathbf{H}_{idx}^{b-1}$ is computed as follows:

$$\mathbf{H}^{b-1} = \text{PDESolve}(\boldsymbol{\alpha}^{b-1}, \mathcal{T}^{b-1}, \mathcal{X}_1, \ldots, \mathcal{X}_d), \tag{78}$$

and

$$idx = (t_{(b-1)*(L-O)+1}, x_{11}, \ldots, x_{d1}), \tag{79}$$

which is the index corresponding to the initial latent state of the $b$-th sub-trajectory in the $(b-1)$-th sub-trajectory.

Next, we define the data likelihood, which models the observed trajectory data $\mathbf{F}$ conditioned on the latent states $\boldsymbol{\alpha}_{\mathbf{F}}^{1:B}$:

$$p_\theta(\mathbf{F} | \boldsymbol{\alpha}_{\mathbf{F}}^{1:B}) = \prod_{b=1}^B p_\theta(\mathbf{F}^b | \boldsymbol{\alpha}_{\mathbf{F}}^b), \tag{80}$$

where each sub-trajectory observation is modeled as a product of likelihoods over the set of observed values $\mathcal{S}^b$:

$$p_\theta(\mathbf{F}^b | \boldsymbol{\alpha}_{\mathbf{F}}^b) = \prod_{s \in \mathcal{S}^b} p_\theta(\mathbf{F}_s^b | \boldsymbol{\alpha}_{\mathbf{F}}^b), \tag{81}$$

and the conditional likelihood for each observed value $\mathbf{F}_s^b$ is Gaussian:

$$p_\theta(\mathbf{F}_s^b | \boldsymbol{\alpha}_{\mathbf{F}}^b) = \mathcal{N}(\mathbf{F}_s^b | \hat{\mathbf{F}}_s^b, \sigma_d^2 I). \tag{82}$$

Here, $\hat{\mathbf{F}}^b$ is the predicted value, computed as:

$$\hat{\mathbf{F}}^b = \text{MLP}(\text{PDESolve}(\boldsymbol{\alpha}_{\mathbf{F}}^b, \mathcal{T}^b, \mathcal{X}_1, \ldots, \mathcal{X}_d)), \tag{83}$$

where $\mathcal{S}^b$ represents the indices of the observed values for the $b$-th sub-trajectory, and $\sigma_d$ controls the variance of the observation noise.

Given a set of parameters $\theta$, the latent states $\boldsymbol{\alpha}_{\mathbf{F}}^{1:B}$ for each trajectory $\mathbf{F}$ are estimated using Maximum-A-Posteriori (MAP) estimation:

$$\hat{\boldsymbol{\alpha}}_{\mathbf{F}}^{1:B} = \underset{\boldsymbol{\alpha}_{\mathbf{F}}^{1:B}}{\arg\max} \, p_\theta(\boldsymbol{\alpha}_{\mathbf{F}}^{1:B} | \mathbf{F}) = \underset{\boldsymbol{\alpha}_{\mathbf{F}}^{1:B}}{\arg\max} \log p_\theta(\boldsymbol{\alpha}_{\mathbf{F}}^{1:B} | \mathbf{F}). \tag{84}$$

The parameters $\theta$ are then estimated by maximizing the posterior across all trajectories in the dataset $\mathcal{D}$:

$$\hat{\theta} = \underset{\theta}{\arg\max} \sum_{\mathbf{F} \in \mathcal{D}} \underset{\boldsymbol{\alpha}_{\mathbf{F}}^{1:B}}{\max} \log p_\theta(\boldsymbol{\alpha}_{\mathbf{F}}^{1:B} | \mathbf{F}), \tag{85}$$

which, using Bayes' Law, can be rewritten as:

$$\hat{\theta} = \underset{\theta}{\arg\max} \sum_{\mathbf{F} \in \mathcal{D}} \underset{\boldsymbol{\alpha}_{\mathbf{F}}^{1:B}}{\max} (\log p_\theta(\mathbf{F} | \boldsymbol{\alpha}_{\mathbf{F}}^{1:B}) + \log p_\theta(\boldsymbol{\alpha}_{\mathbf{F}}^{1:B})). \tag{86}$$

During training, $\theta$ and $\boldsymbol{\alpha}_{\mathbf{F}}^{1:B}$ are jointly optimized. The objective, based on Equation 86, is given by:

$$\hat{\theta}, \hat{\boldsymbol{\alpha}}_{\mathbf{F}}^{1:B} = \underset{\theta, \{\boldsymbol{\alpha}_{\mathbf{F}}^{1:B}\}_{\mathbf{F} \in \mathcal{D}}}{\arg\max} \sum_{\mathbf{F} \in \mathcal{D}} \left( \sum_{b=1}^{B} \sum_{s \in \mathcal{S}^b} \log \mathcal{N}(\mathbf{F}_s^b | \hat{\mathbf{F}}_s^b, \sigma_d^2 I) \right.$$
$$\left. + \log \mathcal{N}(\boldsymbol{\alpha}_{\mathbf{F}}^1 | 0, \sigma_i^2 I) + \sum_{b=2}^{B} \log \mathcal{N}(\boldsymbol{\alpha}_{\mathbf{F}}^b | \mathbf{H}_{idx}^{b-1}, \sigma_c^2 I) \right), \tag{87}$$

which is equivalent to minimizing the following loss function:

$$\hat{\theta}, \hat{\boldsymbol{\alpha}}_{\mathbf{F}}^{1:B} = \underset{\theta, \{\boldsymbol{\alpha}_{\mathbf{F}}^{1:B}\}_{\mathbf{F} \in \mathcal{D}}}{\arg\min} \sum_{\mathbf{F} \in \mathcal{D}} \left( \sum_{b=1}^{B} \sum_{s \in \mathcal{S}^b} \frac{1}{\sigma_{\text{data}}^2} \mathcal{L}(\mathbf{F}_s^b, \hat{\mathbf{F}}_s^b) + \frac{1}{\sigma_0^2} \mathcal{L}(\boldsymbol{\alpha}_{\mathbf{F}}^1, 0) + \sum_{b=2}^{B} \frac{1}{\sigma_c^2} \mathcal{L}(\boldsymbol{\alpha}_{\mathbf{F}}^b, \mathbf{H}_{idx}^{b-1}) \right), \tag{88}$$

where $\mathcal{L}(x, y)$ represents the L2 loss function.

During testing, the initial latent states $\boldsymbol{\alpha}_{\mathbf{F}}^{1:B}$ for a given trajectory $\mathbf{F}$ are estimated by minimizing the following loss, as derived from Equation 84:

$$\hat{\boldsymbol{\alpha}}_{\mathbf{F}}^{1:B} = \underset{\boldsymbol{\alpha}_{\mathbf{F}}^{1:B}}{\arg\min} \sum_{b=1}^{B} \sum_{s \in \mathcal{S}^b} \frac{1}{\sigma_d^2} \mathcal{L}(\mathbf{F}_s^b, \hat{\mathbf{F}}_s^b) + \frac{1}{\sigma_i^2} \mathcal{L}(\boldsymbol{\alpha}_{\mathbf{F}}^1, 0) + \sum_{b=2}^{B} \frac{1}{\sigma_c^2} \mathcal{L}(\boldsymbol{\alpha}_{\mathbf{F}}^b, \mathbf{H}_{idx}^{b-1}), \tag{89}$$

which corresponds to a MAP estimation for the latent states given the observed trajectory $\mathbf{F}$ and the model parameters $\theta$.

# D EXPERIMENTAL SETUP

## D.1 DATASETS

**Advection Equation.** The governing equation for the advection process is given by the following PDE:

$$\partial_t u(t, x) + \beta \partial_x u(t, x) = 0, \quad x \in (0, 1), t \in (0, 2], \tag{90}$$

$$u(0, x) = u_0(x), x \in (0, 1), \tag{91}$$

where $\beta = 0.4$ is a constant advection speed. We consider periodic boundary conditions and the initial condition is specified as a superposition of sinusoidal waves:

$$u_0(x) = \sum_{k_i = k_1, \ldots, k_N} A_i \sin(k_i x + \phi_i), \tag{92}$$

where $k_i = 2\pi \{n_i\} / L_x$ are wave numbers whose $\{n_i\}$ are integer numbers selected randomly in $[1, n_{\max}]$, and $L_x$ is the calculation domain size. The coefficients $A_i$ are randomly chosen from a uniform distribution over the interval $[0, 1]$, and the phase terms $\phi_i$ are selected randomly from the interval $(0, 2\pi)$. We set $k_{\max} = 8$ and $N = 2$. The dataset further incorporates random transformations to the initial condition. Specifically, with a probability of 10%, the absolute value operation is applied with a random sign, and a window function is also applied randomly. Due to computational constraints, we reduced the size of the computational domain to one-quarter of its original extent. The resulting dataset contains 1,800 training samples and 200 test samples.

**Burgers Equation.** The Burgers equation is a PDE modeling the non-linear behavior and diffusion process in fluid dynamics as:

$$\partial_t u(t, x) + \partial_x \left( u^2(t, x)/2 \right) = \nu/\pi \partial_{xx} u(t, x), \quad x \in (0, 1), t \in (0, 2], \tag{93}$$

$$u(0, x) = u_0(x), x \in (0, 1). \tag{94}$$

where $\nu = 0.01$ is the diffusion coefficient and we set 0.01. Similar to the case of the advection equation, periodic boundary conditions are applied, and Equation 92 is used as the initial condition. Due to computational constraints, we reduced the size of the computational domain to one-quarter of its original extent. The resulting dataset contains 1,800 training samples and 200 test samples.

## D.2 BASELINES

To evaluate the effectiveness of Neural PDE, we selected 7 models as baselines:

- **U-Net** (Ronneberger et al., 2015): Originally proposed for image segmentation, U-Net is a convolution-based architecture that has been adapted for spatiotemporal dynamic reconstruction. It processes sparse inputs to produce complete reconstructions (Chai et al., 2020; Takamoto et al., 2022; Li et al., 2024).

- **S³GM** (Li et al., 2024): S³GM employs a Video U-Net model and follows a two-stage approach. It first learns the joint spatiotemporal distribution, then performs reconstruction conditioned on sparse measurements.

- **FNO** (Li et al., 2021): FNO utilizes Fast Fourier Transform to perform convolutions in the spectral domain, enabling efficient learning of mappings between functional spaces.

- **LNO** (Cao et al., 2024): LNO extends the operator learning framework to non-periodic and transient dynamics by leveraging the Laplace transform.

- **DeepONet** (Lu et al., 2021): DeepONet leverages the universal operator approximation theorem to construct a network architecture capable of learning mappings between function spaces.

- **PINN** (Raissi et al., 2019): PINN is a deep learning framework that incorporates physical laws described by partial differential equations into the training process to solve scientific problems from data.

- **MMGNet** (Luo et al., 2024): MMGNet uses implicit neural representations with a separation of spatial and temporal variables to reconstruct continuous physical fields from sparse data.

Following the experimental setup in (Li et al., 2024), we ensure temporal correlation in U-Net, FNO, LNO, and DeepONet by concatenating 20 consecutive time steps along the channel dimension.

## D.3 RESULTS

Figure 7 and Figure 8 illustrate the generalization across tasks and continuity evaluation experiment, respectively.

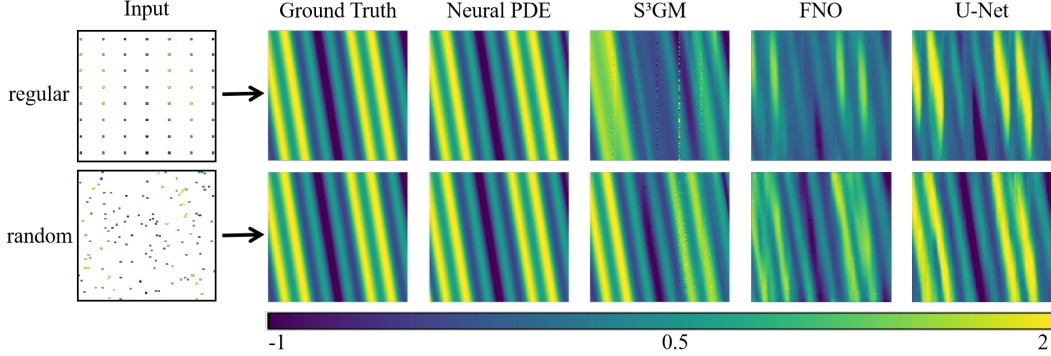

Figure 7: Visualization of generalization across tasks for different models on the regular downsampling task with a 16 downsampling factor and the random downsampling task with a 1% downsampling rate. The models are trained on the random downsampling task with a rate of 2%.

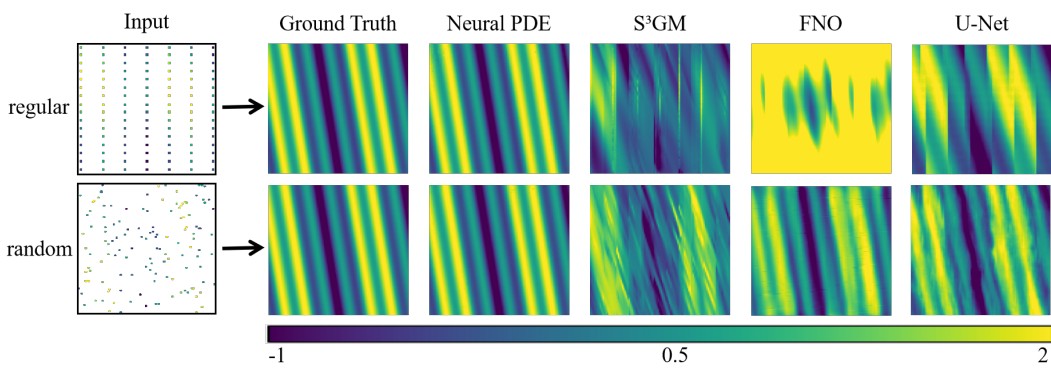

Figure 8: Visualization of continuity evaluation experiment for different models on the regular downsampling task with a 16 downsampling factor and the random downsampling task with a 1% downsampling rate. The models are tested on tasks with a resolution twice that of the training tasks.

