# OpenReview forum: "Continuous Reconstruction of Spatiotemporal Dynamical System with Neural PDE"
_ICLR.cc/2026/Conference — Submitted to ICLR 2026_

### Official Review · Reviewer_4F2c · 2025-10-27

**Soundness:** 3
**Presentation:** 2
**Contribution:** 2
**Rating:** 2
**Confidence:** 5

**Summary:**

This work proposes a super-resolution method that can achieve continuous spatial reconstruction. This problem is important in physical systems, and the method proposed by the author is novel and unique. However, due to insufficient evaluation, it is currently difficult to verify the advantages of the claim. If the author can solve my questions, I will improve the rating.

**Strengths:**

1. The proposed method is novel.
2. The problem being solved is important.

**Weaknesses:**

1. In Line 71, what is path independent? More explanation is needed.
2. In Line 77, the contribution is not written clearly, with excessive jumping, as if there is no continuity of each sentence, which leads to not understanding your core contribution. Moreover, for the three challenges mentioned above, I haven't seen any solutions in the contribution.
3. In Line 82, the underlying equations you mentioned are inaccurate. The paper should not explicitly prove this point or apply specific equations. This description is an overcolaim. Please discuss what pre-training really learns and why pre training is useful.
4. In Line 112, what are Ni and Nd in the form? I can't see the defination of Nd.
5. At the beginning of Methodology, this is a method overview to explain the proposed method, but it cannot correspond to the following subsections because the current method is still difficult to understand?
6. The caption of Figure 2 is too simple to understand the modules in the diagram, and too many complex arrows should be explained.
7. Only two simple 2D PDE systems were selected. Without considering 3D, the above derivation was also conducted in 2D. Can it be extended to 3D？
8. There are many complex equations in PDEBench, such as NSE, and the two chosen are too simple. For sparse observations, this situation often exists in the real world, and it is highly recommended to add real data for verification, as there is a significant gap between real data and simulation data.
9. In Line 449, the generalization here seems to be from more points to fewer points, requiring more experimental details, including Table 2 above which is difficult to understand. If so, can we generalize from fewer points to more points?
10. The major concern. In Line 463. The verification of continuity is insufficient. The first word in the paper title is continuous, but this verification is placed in the last small paragraph, which may lead to an overcolaim. Another important issue is whether the input sparse point grid is on 128 or 256 when testing continuity. If it is on 256, it is unreasonable because continuity should reflect that any point in space can be obtained. If 256 is used, there will be exposed information. It appears that the maximum 16 times super-resolution is downsampled on a grid of 256. If the author believes that the current experiment of the task is reasonable, then the author should demonstrate through other experiments that what the model learns is continuous, rather than super-resolution at different resolutions.

**Questions:**

See the weaknesses.

---

### Official Review · Reviewer_839V · 2025-10-29

**Soundness:** 2
**Presentation:** 2
**Contribution:** 2
**Rating:** 2
**Confidence:** 4

**Summary:**

This paper presents a framework for continuous reconstruction of spatiotemporal dynamics by extending Neural ODEs to each dimension. This pipeline is within a generative pretraining framework. The proposed method directly models higher-order partial derivatives of hidden states to preserve spatial-temporal continuity and ensure path independence in reconstruction. The experiments have shown the superiority of the proposed method on advection and Burgers equations.


Contribution:

- The majority of existing works are doing discrete reconstruction. Doing continuous reconstruction for spatiotemporal dynamics is critical.

- This paper presents a method that combines fitting of partial derivatives, overlapping multiple shooting, continuity loss, etc.

**Strengths:**

- This topic is important. The majority of research works focus on discrete learning instead of continuous learning. Thus, working on continuous reconstruction is critical.

- The experiments have shown the superiority of the proposed framework on advection and Burgers equations.

**Weaknesses:**

- The topic of continuous space-time modeling is not new. There are plenty of research works around this topic [1-3,5]. The authors need to have a richer discussion of the related works and how to differentiate between the proposed work and the literature. Also, PINN and FNO are both space-time continuous methods as long as you have time and spatial coordinates as inputs. The authors may consider rephrasing the statement in Related Work on Page 2.

- The clarity and presentation can be improved. My first impression of this paper is that the authors are not quite familiar with the recent work in this area, specifically around continuous reconstruction and spatiotemporal modeling, since the related work part is well-written, and the naming of this model is quite vague. See more details in Questions.

- The experiments should be improved. As a method paper, I expect to see more rigorous evaluations. For example, the authors might consider many other PDE scenarios, such as 2D NS and 3D NS in the PDEBench. The authors might also consider some other PDE Benchmarks, such as the Well [4] and the datasets from [2].

---

**Refs:**

[1] Iakovlev, et al. "Learning space-time continuous latent neural PDEs from partially observed states." Advances in Neural Information Processing Systems 36 (2023): 26372-26395.

[2] Luo, et al. "Continuous field reconstruction from sparse observations with implicit neural networks." arXiv preprint arXiv:2401.11611 (2024).

[3] Li, et al. "Learning spatiotemporal dynamics with a pretrained generative model." Nature Machine Intelligence 6.12 (2024): 1566-1579.

[4] Ohana, et al. "The well: a large-scale collection of diverse physics simulations for machine learning." Advances in Neural Information Processing Systems 37 (2024): 44989-45037.

[5] Li, et al. "Physics-aligned field reconstruction with diffusion bridge." The Thirteenth International Conference on Learning Representations. 2025.

**Questions:**

- Why do you have to name it Neural PDEs? Neural PDEs is quite a vague name, i.e., many models can be called Neural PDEs (e.g., neural operators), or it sometimes refers to one type of models that use neural networks for PDEs. I would suggest a more specific name.

- What is path independence? Could you give more clarification on this?

- What is the difference between the proposed multiple shooting scheme and the use of overlapping time windows during training? It seems that the authors may be referring to the same concept under different terminology

- On Page 2, ``We propose … into overlapping sub-trajectories to enhances training efficiency and stability.’’ “enhances” should be “enhance”.

---

### Official Review · Reviewer_K9hg · 2025-10-31

**Soundness:** 2
**Presentation:** 3
**Contribution:** 2
**Rating:** 2
**Confidence:** 4

**Summary:**

This paper proposes Neural PDE, which extends Neural ODEs to model spatiotemporal dynamical systems continuously in both space and time. The method enforces path independence by introducing a PDE formulation and integrates a generative pretraining scheme with an auto-decoder to infer latent initial conditions. The authors also propose an “overlapping multiple-shooting” strategy and a continuity loss for stable training. Experiments are conducted on the 1D Advection and Burgers equations from PDEBench.

**Strengths:**

- Clear and rigorous mathematical formulation of a “Neural PDE” concept.
- Attempt to connect ODE-based modeling with PDE operator learning.
- Writing quality and organization are high.

**Weaknesses:**

- Weak results: only 1D Advection and Burgers equations are used; no 2D/3D or real physical systems.
- Weak baselines: comparisons exclude recent operator networks (e.g., PINO, PERCNN).
- Limited novelty: the “Neural PDE” mainly reformulates path independence by integrating existing ODE concepts.
- No real-world or high-dimensional validation: results cannot demonstrate scalability or physical relevance.
- Claims of SOTA are unconvincing given simple examples and simplistic baselines.

**Questions:**

- Can the authors show results on 2D/3D PDEs (e.g., Navier-Stokes) to justify the claimed generality?
- How does “Neural PDE” differ mathematically from simply enforcing symmetry of mixed derivatives in a Neural ODE framework?
- Could the authors include modern operator baselines (e.g., P2C2Net, PINO, PERCNN) for a fairer comparison?
- How sensitive is the method to latent dimension and sub-trajectory length?

---

### Official Review · Reviewer_Wa8L · 2025-10-31

**Soundness:** 1
**Presentation:** 1
**Contribution:** 1
**Rating:** 2
**Confidence:** 3

**Summary:**

The paper proposes Neural PDE, a continuous spatiotemporal modeling framework that extends Neural ODEs to capture both spatial and temporal dynamics via PDE formulation. It introduces a generative pretraining scheme for latent initialization and an overlapping multiple shooting strategy for stable training. Experiments on PDE benchmarks (advection, Burgers) show strong performance, especially under sparse observations.

**Strengths:**

Proposes a continuous spatiotemporal formulation with effective training via overlapping multiple shooting, achieving strong results under sparse data conditions.

**Weaknesses:**

* Clarity of Research Objective:
The paper needs to clarify its main focus. It is currently unclear whether the goal is to extend Neural ODEs to spatiotemporal settings or to develop a deep learning solver specifically for PDEs. The motivation and positioning are ambiguous. All experiments are conducted on PDE-related data, which makes the earlier emphasis on Neural ODEs confusing and somewhat inconsistent with the rest of the paper.

* Theoretical Validity of Proposition 3.2:
Does Proposition 3.2 provide a sufficient condition? Equations (7) and (8) indicate that satisfying these conditions yields a conservative vector field, which constitutes a necessary condition. However, it is not clear whether these equations are also sufficient. If not, the proposed method may restrict the representational space of possible vector fields, potentially limiting its expressiveness rather than enhancing it.

* Ambiguity in Proposition 3.3:
The statement of Proposition 3.3 is too vague and lacks formal rigor. It should be more precisely formulated to make the theoretical argument convincing.

* Applicability and Limitations of Path Independence Constraint:
Real-world time series and PDE systems often exhibit non-conservative dynamics (e.g., heat loss, viscosity, source terms, nonlinear dissipation). Enforcing path independence in such cases may reduce the model’s expressive power and fail to capture essential non-conservative behaviors. Therefore, the authors should clearly specify the range or types of problems where the proposed model is expected to perform well, and where it may not be suitable.

**Questions:**

It would be helpful to report and compare training time and peak memory usage against baseline methods.

---

### Meta-Review · Area_Chair_3s3x · 2025-12-30

**Summary:**

This work proposed Neural PDE, a continuous spatiotemporal modeling framework that extends Neural ODEs to capture both spatial and temporal dynamics via PDE. It integrated a generative pretraining scheme with an auto-decoder to infer latent initial conditions. Evaluations on two benchmarks demonstrated the good performance of the proposed approach.


Strength:
1. It introduces a new concept called Neural PDE, which is an interesting research topic.

2. Evaluation results on two benchmarks demonstrate the good performance of the proposed method.

Limitations:
1. The technical novelty is limited. The “Neural PDE” mainly reformulates path independence by integrating existing ODE concepts.
2. It needs to compare with more baselines, such as P2C2Net, PINO, and PERCNN. It would be great to do more experiments to report and compare training time and peak memory usage against baseline methods.
3. It may need to clarify Proposition 3.3 so that readers can easily understand.

In sum, the novelty of this work is limited. The authors did not address the concerns raised by reviewers. Hence, I propose to reject this work.

**Reviewer Concerns:**

The authors did not address any concerns from reviewers.

**Reviewer Scores:**

The reviewers will not raise scores since The authors did not respond to their comments

---

### Decision · Program_Chairs · 2026-01-26

Reject